# Synaptic proteins promote calcium-triggered fast transition from point contact to full fusion

Jiajie Diao[1†], Patricia Grob[2†], Daniel J Cipriano[1], Minjoung Kyoung[1‡], Yunxiang Zhang[1], Sachi Shah[1], Amie Nguyen[1], Mark Padolina[1], Ankita Srivastava[1], Marija Vrljic[1], Ankita Shah[1], Eva Nogales[2,3], Steven Chu[4], Axel T Brunger[1]*

[1]Departments of Molecular and Cellular Physiology, Neurology and Neurological Sciences, Structural Biology, Photon Science and Howard Hughes Medical Institute, Stanford University, Stanford, USA; [2]Department of Molecular and Cell Biology and Howard Hughes Medical Institute, University of California at Berkeley, Berkeley, USA; [3]Life Sciences Division, Lawrence Berkeley National Laboratory, Berkeley, USA; [4]Formerly Lawrence Berkeley National Laboratory, and Departments of Physics and Molecular and Cell Biology, University of California at Berkeley, Berkeley, USA

*For correspondence: brunger@stanford.edu

‡Present address: Department of Chemistry and Biochemistry, University of Maryland, Baltimore County, USA

†These authors contributed equally to this work

**Abstract** The molecular underpinnings of synaptic vesicle fusion for fast neurotransmitter release are still unclear. Here, we used a single vesicle–vesicle system with reconstituted SNARE and synaptotagmin-1 proteoliposomes to decipher the temporal sequence of membrane states upon $Ca^{2+}$-injection at 250–500 µM on a 100-ms timescale. Furthermore, detailed membrane morphologies were imaged with cryo-electron microscopy before and after $Ca^{2+}$-injection. We discovered a heterogeneous network of immediate and delayed fusion pathways. Remarkably, all instances of $Ca^{2+}$-triggered immediate fusion started from a membrane–membrane point-contact and proceeded to complete fusion without discernible hemifusion intermediates. In contrast, pathways that involved a stable hemifusion diaphragm only resulted in fusion after many seconds, if at all. When complexin was included, the $Ca^{2+}$-triggered fusion network shifted towards the immediate pathway, effectively synchronizing fusion, especially at lower $Ca^{2+}$-concentration. Synaptic proteins may have evolved to select this immediate pathway out of a heterogeneous network of possible membrane fusion pathways.

## Introduction

There are many examples where protein-mediated membrane fusion plays a key role, such as entry of enveloped viruses, fertilization, development, carcinogenesis, intracellular trafficking, secretion, and neurotransmitter release (*Rothman, 1994*; *Jahn et al., 2003*; *Jahn and Scheller, 2006*; *Harrison, 2008*; *Rizo and Rosenmund, 2008*; *Moreau et al., 2011*). Among these biological processes, synaptic vesicle fusion is unique in that it is both regulated and fast, in the order of a millisecond (*Meinrenken et al., 2003*). Upon an action potential, at most one synaptic vesicle undergoes exocytosis in a synapse among the readily releasable pool (*Dobrunz and Stevens, 1997*), that is, only a small subset of docked synaptic vesicles undergo fusion. How synaptic proteins can achieve fast fusion in such a precisely regulated and $Ca^{2+}$-dependent fashion is a matter of intense investigation and considerable controversy (*Sørensen, 2009*; *Collins et al., 2012*). In this context, reductionist in vitro systems play an important role in uncovering the mechanism of action since they allow manipulations and observations not possible in vivo, and they establish if a particular subset of factors represents a minimal system for promoting fast $Ca^{2+}$-triggered release.

**eLife digest** The central nervous system relies on electrical signals travelling along neurons and through synapses at high speeds. Signals often have to pass between two neurons, or from a neuron to a muscle fiber, and the nervous system relies on a process called membrane fusion to ensure that the neurotransmitter molecules that carry the signal across the synapses are released as quickly as possible. Membrane fusion is an important process in many areas of biology, including intracellular transport and fertilization, but it occurs much faster (millisecond time scale) in the nervous system than anywhere else in the body. The reasons for this have long been a mystery, although calcium ions are known to trigger the fusion process.

The fusion of two biological membranes is similar in many regards to the way that small soap bubbles merge together to form large bubbles. Just as soap bubbles can form a variety of discernible intermediate structures when they merge, so can biological membranes. This means that it is possible to produce a so-called hemifusion intermediate in which the outer layers of the membranes have merged, but the inner layers have not, so it is not possible for anything—such as serotonin, dopamine and other neurotransmitter molecules—to transfer from one membrane to the other.

Diao et al. have used a combination of advanced optical imaging and cryogenic electron microscopy to explore membrane fusion between synthetic membranes that contained reconstituted synaptic proteins, including synaptotagmin and a family of protein receptors called SNAREs. When calcium ions were injected into the synthetic system, the basic characteristics of neurotransmitter release—such as membrane fusion on a millisecond time scale—was observed. Contrary to some theories of membrane fusion, the fastest fusion events did not begin or proceed via a discernible hemifusion intermediate state. Rather, these events proceeded from a 'point contact' state in which the membranes were close to each other (just 1–5 nm apart) without being fused, and were ready to undergo fast fusion once the calcium ions had been injected. And when Diao et al. introduced a protein called complexin, which is known to be important for fast neurotransmitter release in vivo, they observed more immediate fusion events and fewer events that involved a hemifusion intermediate.

With a synthetic system it is possible to perform experiments that are currently not possible with live neurons, and this has allowed Diao et al. to clarify the roles of the individual components in the process of membrane fusion, and could prove useful in efforts to develop novel therapeutic treatments to combat neurological disorders.

Protein-mediated fusion between biological membranes is thought to proceed from a pre-fusion contact to a fusion pore through one or more intermediates (*Chernomordik and Kozlov, 2008*; *Jackson and Chapman, 2008*). A hemifusion intermediate is defined as a membrane state where the outer leaflets of the two juxtaposed membranes exchange lipids; examples include an hourglass shaped 'stalk' and an elongated hemifusion diaphragm. Investigations of both protein-free and protein-mediated liposome fusion in model systems, along with macroscopic continuum models and atomistic computer simulations, revealed at least two pathways: a direct pathway from pre-fusion contact to complete fusion pore via a stalk, and the 'classical' pathway with a hemifusion diaphragm intermediate between the stalk and the fusion pore. The energetics and kinetics of these processes are dependent on many factors, including lipid composition and membrane curvature. Model calculations suggested that a hemifusion diaphragm is an energetically relatively stable, long-lived intermediate, and that the direct pathway from stalk to fusion pore may be faster (*Kuzmin et al., 2001*).

Several key proteins cooperate in $Ca^{2+}$-triggered synaptic vesicle fusion, including the soluble N-ethylmaleimide-sensitive factor attachment protein receptors (SNAREs) (*Söllner et al., 1993*; *Sutton et al., 1998*; *Jahn and Scheller, 2006*), the $Ca^{2+}$ sensor synaptotagmin 1, the activator/inhibitor complexin, and SM proteins (*Rizo and Rosenmund, 2008*; *Südhof and Rothman, 2009*). Exocytosis can proceed by full-collapse fusion, where the opening of a small pore continues to expand to a large pore, and 'kiss-and-run', where a small pore opens transiently but closes again as the vesicle stays associated with the plasma membrane (*Zhang et al., 2009*). There are two macroscopic models for fusion pore formation: lipid-lined where the synaptic proteins act as scaffolds for pore formation and

protein-lined where the transmembrane domains of some of the synaptic proteins play a more active role by lining the pore (*Jackson and Chapman, 2006*; *Ngatchou et al., 2010*). However, atomistic computer simulations suggested that the molecular mechanism of pore formation is likely more complex than implied by simple macroscopic models (*Risselada et al., 2011*; *Lindau et al., 2012*).

SNARE proteins can promote membrane fusion via hemifusion intermediates, as inferred from lipid-mixing studies with proteoliposomes where fluorescent labels measure the exchange of components between lipid bilayers (*Weber et al., 1998*; *Reese and Mayer, 2005*; *Xu et al., 2005*; *Schaub et al., 2006*; *Liu et al., 2008*). However, it is content mixing, that is, the diffusion between compartments of small molecules with size comparable or slightly larger to that of neurotransmitter molecules that is the correlate for neurotransmitter release. Several previous studies implied that content mixing is not necessarily correlated with lipid mixing for biological membrane fusion: neuronal SNAREs alone do not produce much content mixing, although they readily induce lipid mixing (*Bowen et al., 2004*; *Kyoung et al., 2011*), influenza virus-induced fusion content mixing occurs seconds after initial lipid mixing (*Floyd et al., 2008*), and content mixing occurs with minutes delay after lipid mixing in vacuolar fusion (*Jun and Wickner, 2007*). Even inner leaflet mixing can occur without content mixing when fusion is induced by DNA-zippering (*Chan et al., 2009*), although this intriguing observation will require verification with synaptic-protein induced fusion. The results presented in this work reveal an even more striking example of the difference between content mixing and lipid mixing for $Ca^{2+}$-triggered fusion with synaptic proteins.

Which pathway is used for fast synchronous neurotransmitter release? Is it the classical hemifusion diaphragm pathway or a more direct pathway? We investigated this fundamentally important question with our recently developed optical microscopy method that simultaneously monitors the temporal sequence of both content and lipid exchange upon $Ca^{2+}$-triggering between single pairs of donor and acceptor vesicles on a 100-ms time scale (*Kyoung et al., 2011*; *Kyoung et al., 2012*). Donor and acceptor vesicles mimicked synaptic vesicles and the plasma membrane, respectively. Our system can discriminate between docking, hemifusion, and complete fusion. After improvements in optical instrumentation as well as protein expression and purification, here we achieved a $Ca^{2+}$ sensitivity in the 250–500 μM range. This $Ca^{2+}$ range is reasonably close to the physiological concentration range (starting at 10 μM and saturating at several 100 μM) (*Heidelberger et al., 1994*) and comparable to other recent in vitro experiments (*Wang et al., 2011*; *Hernandez et al., 2012*). Our system mimics a stepwise $Ca^{2+}$ concentration increase that acts on the readily-releasable pool of primed synaptic vesicles, and it is thus reminiscent to experiments with live neurons using photolysis of caged $Ca^{2+}$ compounds (*Schneggenburger and Neher, 2000*; *Sun et al., 2007*). We anticipate that a different $Ca^{2+}$ delivery technique for our system will allow us to mimic the transient $Ca^{2+}$ pulses (or trains of such pulses) that are more typical for physiological neurotransmitter release.

We combined our optical microscopy experiments with cryo-electron microscopy (cryo-EM) image analysis of mixtures of the same types of donor/acceptor vesicles before and after $Ca^{2+}$-addition. We observed a variety of morphologies of membrane-membrane interfaces, as well as changes in the distribution of membrane–membrane interfaces upon $Ca^{2+}$ addition. Taken together, we deciphered the pathway that allows synaptic vesicles to fuse immediately upon $Ca^{2+}$-triggering. We found that all fast (immediate) fusion events start from a hemifusion-free state and proceed to full fusion upon $Ca^{2+}$ injection without discernible hemifusion intermediates. In contrast, stable, initially hemifused states are slow to fuse, if at all. Moreover, we found that complexin dramatically increases the number of immediate fusion events, with a more pronounced effect at lower $Ca^{2+}$ concentration (250 μM).

## Results

### $Ca^{2+}$-triggered fusion with SNAREs and synaptotagmin 1 at 500 μM

We modified our original single vesicle–vesicle microscopy system (*Kyoung et al., 2011*; *Kyoung et al., 2012*) and improved protein quality in order to enable studies at 250–500 μM $Ca^{2+}$-concentration (for details, see 'Materials and methods'). As before, we used membrane compositions and protein number densities that mimic synaptic vesicles and the plasma membrane, respectively, and reconstituted full-length synaptotagmin 1 together with synaptobrevin (also referred to as VAMP, Vesicle Associated Membrane Protein), and syntaxin together with SNAP-25 into two separate populations of liposomes, termed donor and acceptor vesicles, respectively. We had previously established the homogeneity of our vesicle preparations (*Kyoung et al., 2011*). Unlabeled acceptor vesicles were tethered

to a PEG-coated glass surface, and the donor vesicles were labeled with self-quenched lipid (DiD) and content (sulforhodamine B) fluorophores (*Figure 1A*). After a defined incubation period our system started from a metastable state of docked vesicles at zero $Ca^{2+}$ concentration. Upon $Ca^{2+}$ injection we monitored 'changes' in membrane state in real-time by visual inspection of individual content and lipid mixing fluorescence intensity time traces and identification of significant jumps in these traces.

Representative examples for $Ca^{2+}$-injection at 500 μM are shown in *Figure 1B* for experiments with full-length SNAREs and full-length synaptotagmin 1. We observed three characteristic changes in membrane state, including instances of immediate fusion (a correlate with synchronous neurotransmitter release), delayed fusion (a correlate with asynchronous release), and hemifusion-only (i.e., without complete fusion) during the observation period of 50 s. Overall, for this very minimal system consisting of only SNAREs and synaptotagmin 1, immediate fusion instances were in the minority, followed by delayed fusion instances, and then hemifusion-only instances (*Figure 1C*). As we had shown previously, $Ca^{2+}$ triggered fusion depended on the presence of both functional SNAREs and synaptotagmin 1, for example, mutation of one of the $Ca^{2+}$ binding sites of synaptotagmin 1 greatly reduced the amount of immediate fusion and disruption of SNARE complex formation prevented both docking and fusion (*Kyoung et al., 2011*).

## Imaging by cryo-electron microscopy

In order to correlate the temporal sequence of changes in membrane state observed by single vesicle–vesicle microscopy with membrane morphology, we imaged mixtures of the same types of donor/acceptor vesicles before (*Figure 2A–C*) and approximately 35 s after addition of $Ca^{2+}$ (*Figure 2D–F*) by cryo-EM. We used the same defined incubation period for the donor/acceptor vesicle mixture prior to $Ca^{2+}$ addition as for the single vesicle–vesicle microscopy experiments (30 min). Many 'point contacts' (see 'Materials and methods' for definitions of all contact and interface types) between donor and acceptor vesicles were observed before $Ca^{2+}$ addition (*Figure 2B*), as well as hemifusion diaphragms (*Figure 2C*). Both point contacts and hemifusion diaphragms are likely between donor and acceptor vesicles since we did not observe interactions between vesicles of the same type in previous cryo-EM experiments (*Kyoung et al., 2011*). After $Ca^{2+}$ addition, besides point contacts and hemifusion diaphragms (*Figure 2D,E*), we also observed a smaller number of 'extended' close contacts without hemifusion, defined as a triple-layered feature extending over more than 10 nm in length (*Figure 2F*). Interestingly, similar extended close contacts were induced between liposomes by denatured Munc18 (*Xu et al., 2011*), and by SNAREs alone when the membrane-proximal layer of the SNARE was disrupted by mutation (*Hernandez et al., 2012*). Since we observed these hemifusion diaphragms and extended close contacts by cryo-EM imaging approximately 35 s after $Ca^{2+}$-injection, such interfaces are expected to be part of 'slow', that is, delayed fusion pathways.

We performed a quantitative analysis of all observed cryo-EM images. As expected from our single-vesicle microscopy fusion experiments with identical vesicle preparations, some fusion occurred after $Ca^{2+}$-addition, shifting the diameter distribution of vesicles observed in the cryo-EM images towards larger vesicles than before $Ca^{2+}$ addition, that is, there are fewer small vesicles and more larger vesicles than before $Ca^{2+}$-addition (*Figure 2G*). The mean diameter increased from 61.3 ± 18.6 (s.d.) nm to 65.1 ± 23.9 (s.d.) nm (a significant increase with 97% confidence from a Student's t-test), corresponding to an increase of mean volume from $1.6 \times 10^5$ nm$^3$ to $2.2 \times 10^5$ nm$^3$. Moreover, the distribution of vesicle interfaces changed after $Ca^{2+}$ addition (*Figure 2H*): the percentage of point contacts decreased from 55% to 40%, the percentage of hemifused diaphragms increased from 45% to 49%, and the percentage of extended interfaces jumped from zero to 11%.

We analyzed the image density of point contacts at zero $Ca^{2+}$; two representative examples are shown in *Figure 3A*. The cryo-EM images of most point contacts indicated that membranes are close to each other at the tip of the contact but not forming a membrane stalk (an hourglass shaped connection between two bilayers that involves the merger of outer leaflets) since the observed minimal separation between membranes (ranging from 4.9 to 2.3 nm) was larger than the critical distance for membrane stalk formation (0.9 nm) (*Aeffner et al., 2012*). Interestingly, a similar membrane apposition between liposomes could be induced by the soluble C2AB domain of synaptotagmin 1 and $Ca^{2+}$, but without SNAREs (*Araç et al., 2006*), as well as with SNAREs alone (*Hernandez et al., 2012*); in this context it has been suggested that synaptotagmin may act as a distance regulator (*van den Bogaart et al., 2011b*).

We also analyzed the images density of a mixed interface (*Figure 3B*), which likely represents a transition state from extended contact to hemifusion diaphragm along a particular fusion pathway.

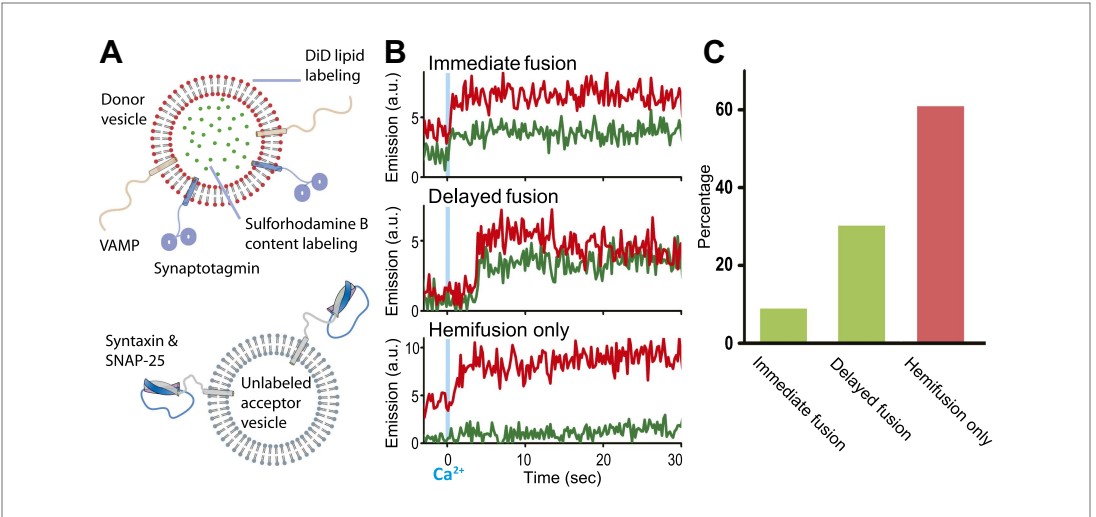

**Figure 1**. Single vesicle–vesicle microscopy for monitoring changes in the membrane and contents upon Ca²⁺-injection. (**A**) Labeling and reconstitution scheme of our single-vesicle lipid/content mixing system. Other factors and proteins can be added to the system. The detailed experimental protocol is described in (***Kyoung et al., 2012***) with modifications as described in 'Materials and methods'. (**B**) Representative real-time fluorescence intensity traces with donor (synaptobrevin and synaptotagmin 1) and acceptor (syntaxin and SNAP-25) vesicles (red/upper traces: lipid dye fluorescence intensity, green/lower traces: content dye fluorescence intensity). Time point 0 indicates the instance of Ca²⁺ injection at 500 μM (blue vertical lines). Shown are instances of 'immediate' fusion, as defined by a content dye fluorescence intensity jump during the first 600 ms time bin upon Ca²⁺-injection along with a simultaneous jump in lipid dye fluorescence intensity, 'delayed' fusion, as defined by a content dye fluorescence intensity jump during a subsequent time bin, and 'hemifusion-only', as defined by a lipid dye fluorescence intensity jump without a content dye fluorescence intensity jump during the observation period of 50 s. For illustration purposes, only 30 s of the observation period are shown since there was no change in the subsequent 20 s. (**C**) Bar graph of the percentage of immediate, delayed fusion, and hemifusion-only events involving SNAREs and synaptotagmin 1. The bar graph was normalized with respect to the number of fluorescent spots that exhibited at least one lipid-mixing event during the observation period. Note, that there are fewer immediate fusion events compared to our previous experiments that were carried out at higher Ca²⁺ concentration (***Kyoung et al., 2011***).

The estimated distance between the two bilayers in the extended contact region is less than 1 nm (***Figure 3B***, middle panel, left half of the image), which is representative for the distances observed for other extended close contacts that are not hemifused.

## Point contacts: starting point for fast Ca²⁺-triggered fusion

Considering the size of synaptic proteins (SNARE complex: approximately 11.6 × 2.6 nm, synaptotagmin C2 domains: approximately 3 × 5 nm), they would have to act at the periphery of many of the observed vesicle interfaces. The point contact is a special case since it is the only interface type where the proteins would be close to a putative transitory stalk to initiate fusion. Indeed, as mentioned above, quantitative analysis of all cryo-EM images revealed that the fraction of point contacts decreased upon Ca²⁺ addition (***Figure 2H***). Inspired by this relative decrease after addition of Ca²⁺, we hypothesized that some of these initial point contacts are the likely starting point for immediate fusion upon Ca²⁺ injection; those that did not fuse might be ready to undergo fusion during subsequent pulses.

Our single vesicle–vesicle microscopy system provided the tool to test if point contacts are the starting points for immediate fusion events. With an extension of our system, we were able to deduce the initial state of the membrane interface (docked or hemifused) at the time of Ca²⁺-injection for each individual fusion event; a change in the single vesicle–vesicle membrane interface during the incubation period would result in a significant difference between the initial lipid dye fluorescence intensity and that after the incubation period (***Figure 4*** and 'Materials and methods'). Remarkably, no significant lipid fluorescence intensity changes (i.e., changes larger than the noise level) were observed during the incubation period for 'all' instances of immediate fusion events (***Figure 4A***), in other

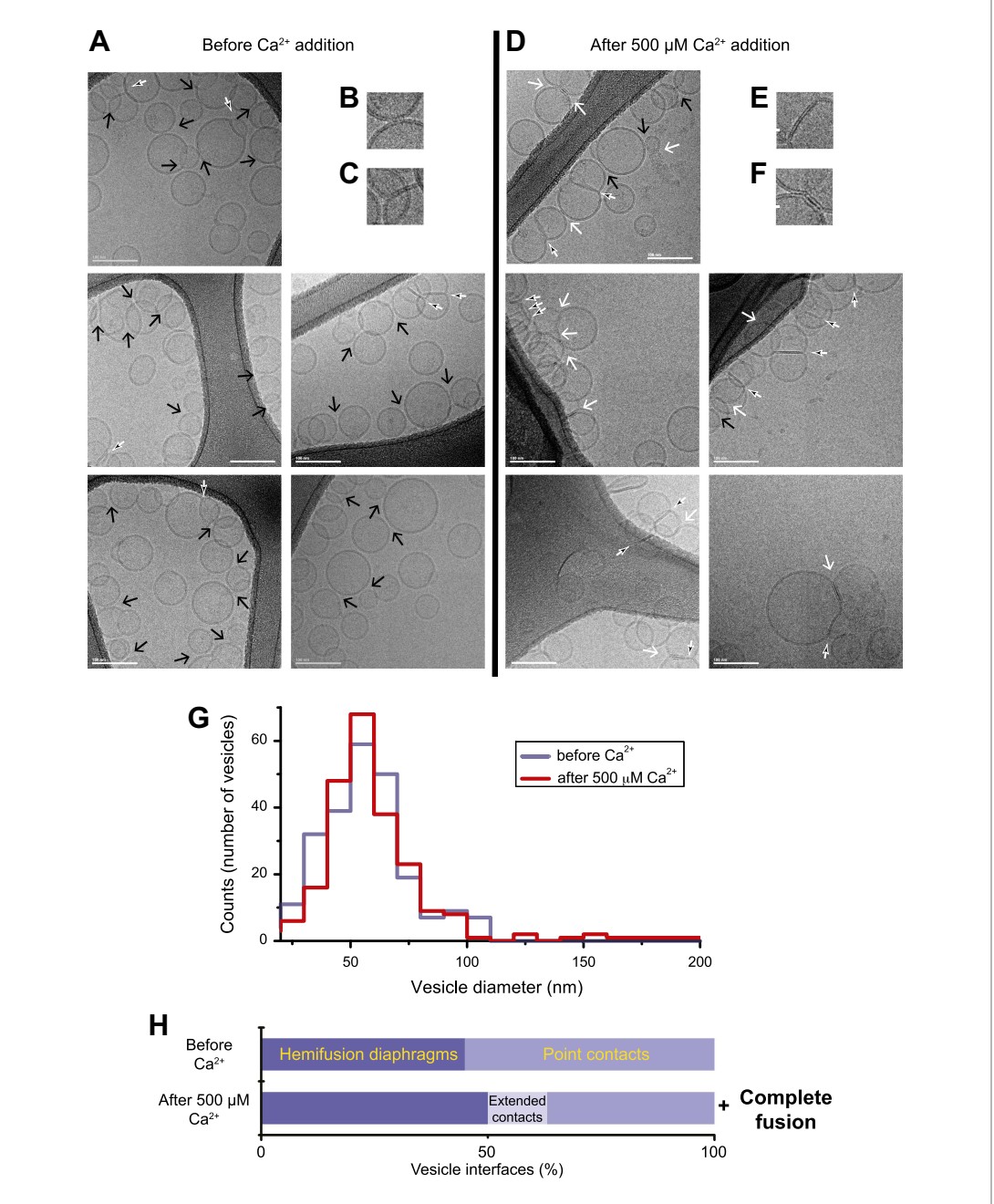

**Figure 2**. Imaging of donor/acceptor interface morphologies by cryo-EM before and after Ca²⁺ addition. (**A–F**) Cryo-EM images of mixtures of donor (synaptobrevin and synaptotagmin 1) and acceptor (syntaxin and SNAP-25) vesicles before (**A–C**) and approximately 35 s after (**D–F**) 500 μM Ca²⁺ addition (panels B, C, E, and F are close-up views). Vesicles were imaged in the holes of the substrate carbon film, visible as the darker areas in the image, in conditions that clearly show the lipid bilayers ('Materials and methods'). The particular images shown in this figure were selected out of total of 16 (before Ca²⁺ addition) and 21 (after Ca²⁺ addition) EM micrographs, respectively, with emphasis on showing point contacts before Ca²⁺ addition and extended interfaces after Ca²⁺ addition in order to illustrate the variety of these particular states. Arrows indicate interfaces between vesicles that are approximately perpendicular to the direction of the projection (see 'Materials and methods' for definitions of all contact and interface types). Large black arrow: point contact (representative close-up in B), small black/white arrow: hemifusion diaphragm (representative close-ups in C and E), and large white arrow: extended close contact (representative close-up in F). Scale bars in A and D are 100 nm, and 20 nm in B, C, E, and F. (**G**) Distribution of vesicle sizes before and after Ca²⁺ addition. Vesicle diameters were calculated from all cryo-EM images in both conditions ('Materials and methods'). (**H**) Bar graph of the percentage of various vesicle interfaces, that is, point contacts, hemifusion

*Figure 2. Continued on next page*

*Figure 2. Continued*

diaphragms, and extended close contacts (including a few instances of mixed, i.e., extended/hemifused, interfaces), normalized with respect to the total number of interfaces observed before and after addition of 500 µM $Ca^{2+}$, respectively ('Materials and methods'). In addition to the changes in the distribution of vesicle interfaces upon $Ca^{2+}$ addition, some amount of complete fusion between vesicles occurred as indicated by the shift of the diameter distribution in panel G towards larger values. Source files of all cryo-EM micrographs used for the quantitative analysis are available in *Figure 2—source data 1*.

The following source data are available for figure 2.
**Source data 1.** Source files for cryo-EM data.

words, these events did not start from a hemifused state. In contrast, $Ca^{2+}$-triggered content mixing rarely occurred during the observation period of 50 s when starting from a stable hemifused state (*Figure 4B*). Since cryo-EM imaging of the membrane interfaces prior to $Ca^{2+}$-addition indicated that there were only two classes, point contacts and hemifusion diaphragms, it follows that all immediate $Ca^{2+}$-triggered fusion events must start from point contacts that are not hemifused. The ability to characterize the initial membrane state and the temporal sequence of events upon $Ca^{2+}$ injection, along with the ability of controlling the constituents, is another illustration of the power of our single vesicle–vesicle microscopy system.

## Modeling of putative protein density in pre-fusion point contact state

As mentioned above, the observed cryo-EM images of point contacts indicate that there is space for synaptic proteins close the tip of the contact between the membranes, although less likely right at the tip. However, few significant densities (greater than a factor of two above noise level) were actually visible near the point contacts, with some notable exceptions (e.g. red arrow in *Figure 5A*). This lack of ubiquitous protein densities could be attributed to the low number of synaptic proteins that are expected to be involved in docking and exocytosis (*van den Bogaart and Jahn, 2011*); proteins could also be overlapping with the interface in projection and be masked by the high-contrast lipids. To support the notion that the few observed instances of significant densities could be related to synaptic proteins, we overlaid two alternative models of the synaptotagmin 1• SNARE complex to the image of a point contact with the highest contrast, assuming that the marked density is related to the globular synaptotagmin C2 domains (*Figure 5B,C*). The apparent lack of density for the SNARE complex agrees with the notion of a partially folded *trans*-SNARE complex that is stabilized by repulsive forces between solvated membranes (*Gao et al., 2012*), as approximately indicated in our models. The single molecule FRET efficiency data of the synaptotagmin 1• SNARE complex indicated multiple binding modes (*Choi et al., 2010*), with the top solution shown in *Figure 5B*. An alterative model of this complex (*Figure 5C*) was based on NMR chemical shift perturbation experiments of the complex between synaptotagmin 1 and the SNARE complex (*Dai et al., 2007*). Both models predict that it is the C2B domain that interacts with the SNARE complex, although the exact interface and orientation between the molecules is different. While approximate, these conceptual models may serve as guide to speculate about the involvement of synaptic proteins in forming the point contact, and setting the stage for $Ca^{2+}$-triggered fusion. For example, the position of synaptotagmin 1 in our models would enable $Ca^{2+}$-triggered membrane juxtaposition (*Araç et al., 2006*) and membrane bending (*Hui et al., 2009*; *McMahon et al., 2010*). Membrane bending or buckling could in turn destabilize the membrane, and in conjunction with full zippering of the SNARE complex, lead to fusion pore opening.

## Effect of complexin upon triggering with 250–500 µM $Ca^{2+}$

The variety of observed fusion pathways induced by the minimal system of full-length SNAREs and full-length synaptotagmin 1 upon $Ca^{2+}$ addition suggest a key role for other synaptic factors to enhance the immediate (fast) fusion pathway from point contact to fusion pore. To test this hypothesis, we included in our system the synaptic protein complexin which has both activating and inhibiting roles for $Ca^{2+}$-evoked and spontaneous neurotransmitter release in vivo, respectively (*Giraudo et al., 2009*; *Maximov et al., 2009*). By design, our in vitro system with SNAREs and synaptotagmin 1 probes the function of complexin for $Ca^{2+}$-evoked release. Strikingly, the content mixing histogram showed more

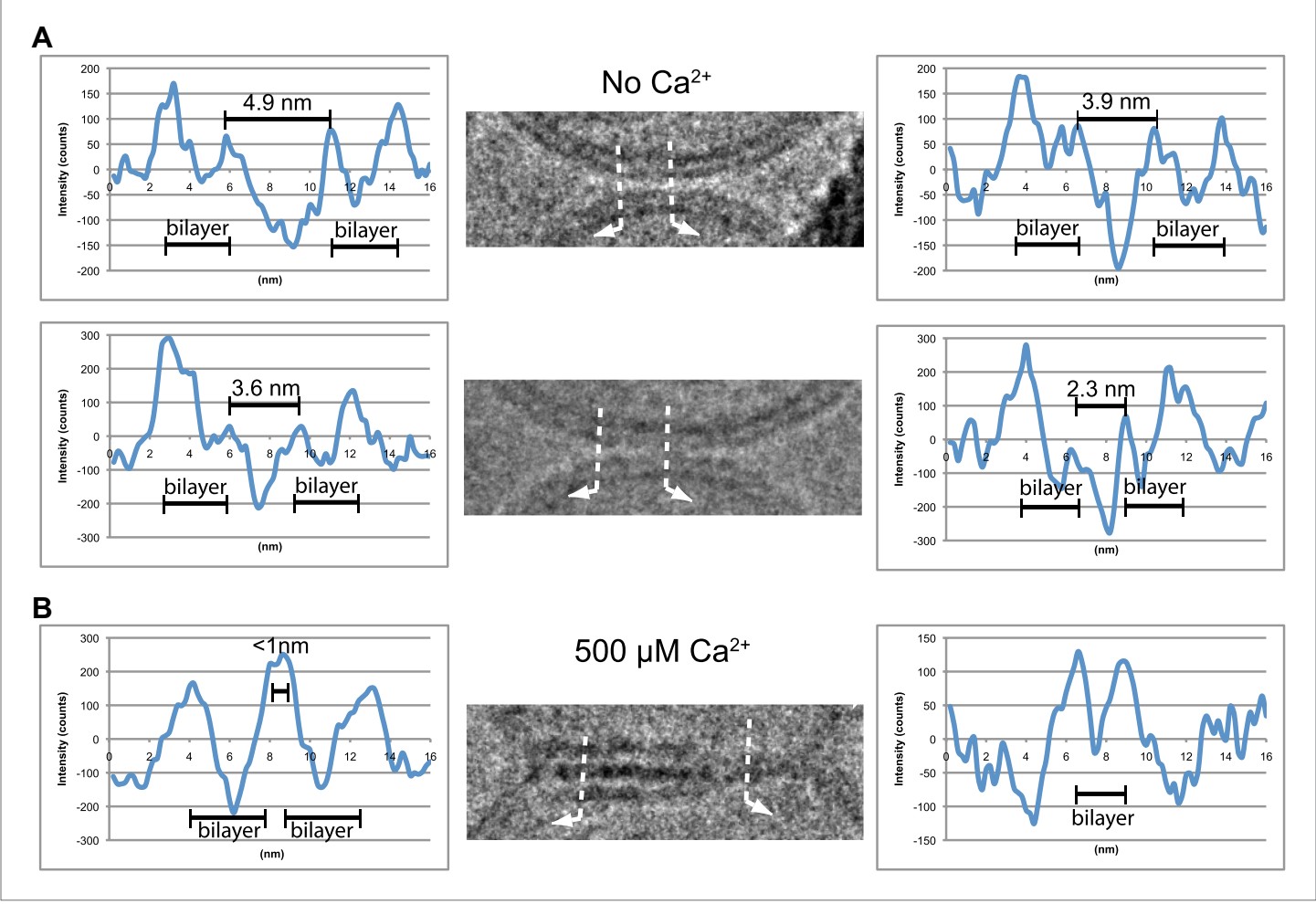

**Figure 3**. Image density profile analysis of selected vesicle–vesicle interfaces. (**A**) Image density profile analysis of two representative point contacts between vesicles at zero Ca²⁺. Dotted lines indicate 2 nm thick sections selected from the cryo-EM images. These sections were used to generate the profiles shown to the left and the right of the close-up views in the center (for details, see 'Materials and methods'). (**B**) Image density profile analysis of a transition from extended close contact to hemifusion diaphragm after addition of 500 μM Ca²⁺. As in panel A, selected sections are indicated by dotted lines, and the corresponding profiles are shown the left and right of the close-up view in the center.

pronounced short-time scale content mixing upon Ca²⁺ injection than in the absence of complexin (***Figure 6A***). The content mixing histogram is related to the probability of fusion vs time, a mimic for evoked postsynaptic currents. Indeed, the content mixing histogram in the presence of complexin could be fitted to a biphasic exponential decay function where the fast fraction with a time constant of 0.36 s accounts for 68% of all events. At lower Ca²⁺ concentration (250 μM), the effect of complexin on the fusion probability time trace became even more pronounced (***Figure 6B***) with 90% in the fast fraction. In contrast, the fusion probability time trace was best fit to a single exponential decay function with a time constant of 3 s in the absence of complexin, which is a significant difference to the content mixing histogram in the presence of complexin.

## Discussion

In this work, we studied the fusion pathways of Ca²⁺-triggered synaptic vesicle fusion with an in vitro system consisting of proteoliposomes reconstituted with synaptic proteins. We used single vesicle–vesicle microscopy to discriminate and quantify the temporal sequence of changes in membrane state (docking, hemifusion, or fusion) on a 100-ms time scale. Our system starts from a meta-stable state of single docked donor/acceptor vesicles that is established by an incubation stage at zero Ca²⁺ concentration. Optical tweezer pulling experiments recently revealed a partially folded

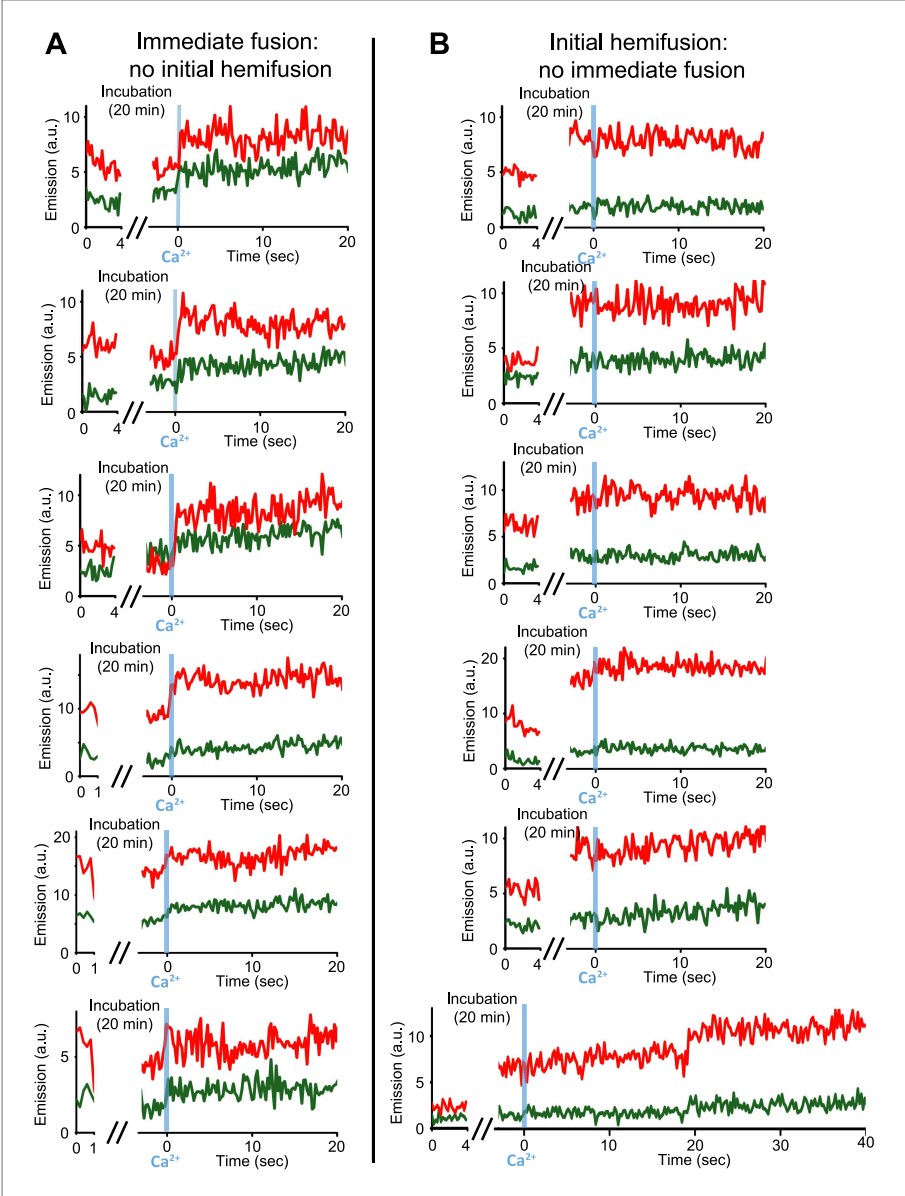

**Figure 4**. Upon Ca²⁺-injection, immediate fusion events start form hemifusion-free point contacts whereas initially hemifused states are slow to fuse. (**A**) Representative real-time fluorescence intensity traces of instances of immediate fusion with imaging of initial florescence intensity levels right after addition of donor (synaptobrevin and synaptotagmin 1) to acceptor vesicles (syntaxin and SNAP-25) (red/upper traces: lipid dye fluorescence intensity, green/lower traces: content dye fluorescence intensity). Time point 0 indicates the instance of Ca²⁺ injection at 500 µM (blue vertical lines). As in *Figure 1*, immediate fusion is defined as a content dye fluorescence intensity jump that occurs during the first 600 ms time bin upon Ca²⁺ injection. In all cases, no significant (i.e., above noise level) lipid fluorescence intensity change was observed during the incubation period, which excludes the possibility of a hemifused initial state before Ca²⁺ triggering. Moreover, 'all' (seven) observed traces with immediate fusion upon Ca²⁺ injection exhibited this behavior. (**B**) Representative real-time fluorescence intensity traces of cases where the membranes are hemifused prior to Ca²⁺-injection. Initial hemifusion was characterized by a significant increase (i.e., above noise level) in only lipid dye fluorescence intensity between the initial recording and at the end of incubation period at 0 Ca²⁺. For approximately 400 fluorescent spots co-localized in both the initial recording and at the end of the incubation period, we observed 51 traces showing (1) an increase in lipid dye fluorescence and (2) no change in content dye fluorescence intensity during the incubation period but before the Ca²⁺ injection. For most of the 51 cases, no content mixing was observed upon Ca²⁺-injection during the observation period of 50 s except for two cases (see one example in the bottom panel) where delayed content mixing occurred at a later time. For illustration purposes only 20 s of the 50 s observation period are shown all traces but one since there was no change in the subsequent 30 s.

**Figure 5**. Modeling of putative protein density in a pre-fusion point contact state. (**A**) Close up view of *Figure 3A*, top middle panel, around a significant density feature (greater than a factor of two above noise level) that is not part of the membranes (red arrow). Only the density features on the left-hand side of the panel were considered since the carbon grid (dark feature in the lower right corner in *Figure 3A*, top middle panel) may have affected protein interactions. (**B**) Overlay of the model of the complex of the C2A–C2B fragment of synaptotagmin 1 (cyan) and the neuronal SNARE complex (synaptobrevin—blue, syntaxin—red, SNAP-25—green) based on 34 single-molecule FRET experiments (*Choi et al., 2010*). The last four C-terminal helical turns of synaptobrevin were modeled as a random coil. (**C**) Overlay of a model of the complex of the C2B domain of synaptotagmin 1 and the neuronal SNARE complex based on proximity between the C2B domain polybasic region and residues D186, D193 of SNAP-25, both of which show sizeable chemical shift perturbations in NMR experiments upon complex formation, among other perturbations (*Dai et al., 2007*). Six C-terminal α-helical turns of synaptobrevin and one α-helical turn of syntaxin were modeled as a random coil. The transmembrane domains were modeled as α-helices, and the linkers as random coils. See 'Materials and methods' for details.

intermediate state of the SNARE complex under external force conditions (*Gao et al., 2012*). This intermediate state likely facilitates the metastable state of interacting and largely unfused vesicles during the incubation stage at zero $Ca^{2+}$ at ambient (25°C) temperature that is the starting point for our system. On a more technical note, the incubation period at zero $Ca^{2+}$ should reduce *cis*-interactions between the C2 domains of synaptotagmin to its own membrane that otherwise might prevent proper *trans*-interactions with acceptor vesicles in assays that use a constant non-zero $Ca^{2+}$-concentration (*Vennekate et al., 2012*).

With improvements to our optical apparatus and protein quality we obtained a $Ca^{2+}$ sensitivity in the 250–500 µM range for our minimal system consisting of SNAREs, synaptotagmin 1, and complexin. This $Ca^{2+}$-range is reasonably close to the physiological concentration range (starting at 10 µM and saturating at several 100 µM) (*Heidelberger et al., 1994*), although it is conceivable that the addition of other synaptic factors will further increase the $Ca^{2+}$ sensitivity. We combined our optical microscopy experiments with cryo-EM imaging of donor/acceptor vesicle mixtures before and after $Ca^{2+}$ addition. Taken together, we obtained for the first time quantitative information about membrane states before and after $Ca^{2+}$ addition, and their temporal changes.

## A striking example that lipid mixing does not imply content mixing

As mentioned in the 'Introduction', lipid mixing (exchange between membranes) is necessary for subsequent content mixing (complete fusion), but it is not sufficient. Moreover, it is content mixing that is relevant for neurotransmitter release, not lipid mixing. As a further demonstration of this principle, complexin had little effect on the lipid mixing histogram upon 500 µM $Ca^{2+}$ injection (*Figure 8A*). In marked contrast, complexin had a large effect on the corresponding content mixing histograms (*Figure 6A*). Likewise, at 250 µM $Ca^2$ the effect of complexin is more pronounced on the content mixing histogram compared to the lipid mixing histogram (compare *Figures 6B and 8B*). This large difference in complexin-induced lipid mixing and content mixing effects greatly amplifies previous observations for biological membrane fusion (*Bowen et al., 2004*; *Jun and Wickner, 2007*; *Floyd et al., 2008*; *Chan et al., 2009*; *Kyoung et al., 2011*). Taken together, these observations, and especially the results presented in this work for $Ca^{2+}$ triggered fusion, call into question conclusions in the literature drawn from lipid-mixing assays when they were not validated by observations related to content mixing. It is thus essential that biological fusion experiments monitor a quantity that is directly related to content mixing in order to distinguish between different membrane states, such as docking, hemifusion, and full fusion.

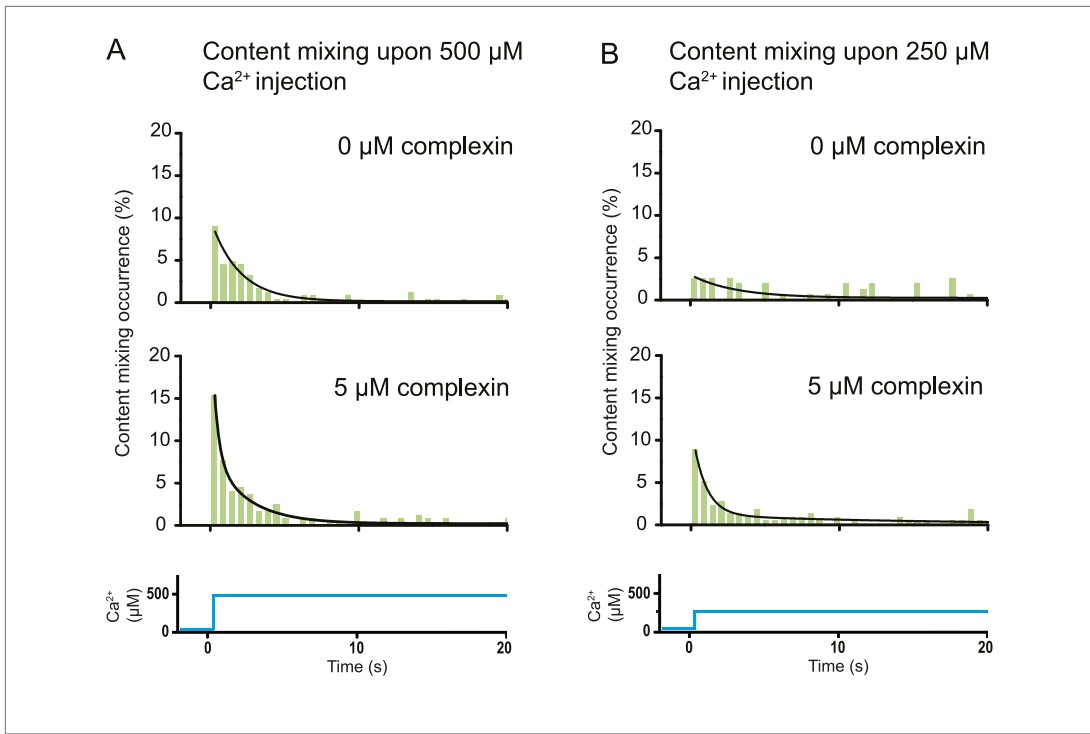

**Figure 6**. Effect of complexin on Ca²⁺-triggered fusion probability. Probability of fusion vs time upon 500 μM (**A**) or 250 μM (**B**) Ca²⁺-injection (i.e., occurrence of Ca²⁺-triggered content-mixing events vs time) with and without 5 μM complexin. The same donor (synaptobrevin and synaptotagmin 1) and acceptor (syntaxin and SNAP-25) vesicles were used as in *Figure 1*. Histograms were normalized to the number of respective fluorescence intensity time traces showing at least one lipid-mixing event during the observation period of 50 s (see 'Materials and methods' for details). For illustration purposes only 20 s of the 50 s observation period are shown since there were few events in the subsequent 30 s; a time binning of 600 ms was used. Black lines are fits to exponential decay functions over the entire 50 secs. For 500 μM Ca²⁺, in the absence of complexin the fitted function is $f(t) = 0.0015 + 0.096e^{-t/1.93}$ and it is 10 times more likely compared to a two-exponential fit, and in the presence of 5 μM complexin the fitted function is $f(t) = 0.0021 + 0.084e^{-t/2.38} + 0.18e^{-t/0.36}$ and it is 10⁷ times more likely compared to a single exponential fit. For 250 μM Ca²⁺, in the absence of complexin the fitted function is $f(t) = 0.0026 + 0.027e^{-t/2.98}$ and it is three times more likely compared to a two-exponential fit, and in the presence of 5 μM complexin the fitted function is $f(t) = 0.0005 + 0.10e^{-t/0.89} + 0.011e^{-t/14.5}$ and it is 10¹⁴ times more likely compared to a single exponential fit.

## Observation of a heterogenous network of fusion pathways

The combination of single vesicle–vesicle microscopy and cryo-EM imaging experiments allowed us to decipher fusion pathways that were induced by the combination of full-length SNAREs, full-length synaptotagmin 1, and, in some experiments, complexin. Before Ca²⁺ addition, the majority (55%) of interacting vesicles formed point contacts, while the remaining vesicle interfaces consisted of hemifusion diaphragms (*Figure 2H*) with SNAREs and synaptotagmin 1. Only a small fraction of vesicles underwent spontaneous (complete) fusion during the incubation period.

Starting from hemifused vesicles, little complete fusion was observed upon Ca²⁺-injection (*Figure 4B*). In contrast, starting from point contacts, we observed a heterogeneous network of both immediate and delayed fusion pathways upon Ca²⁺ injection (*Figure 7*). It is remarkable that the very minimal system of just SNAREs and synaptotagmin 1 exhibited Ca²⁺ triggered activity, including instances of immediate fusion (*Figure 1C*). The delayed fusion pathways proceeded on a second to minute time scale via long-lived extended contact or hemifusion intermediates (*Figure 7*).

## Fast fusion does not begin, or proceed via, stable hemifusion diaphragms

Our experiments revealed the pathway for immediate Ca²⁺-triggered fusion by synaptic proteins (*Figure 7*). It starts from a hemifusion-free point contact and proceeds to full fusion without discernible

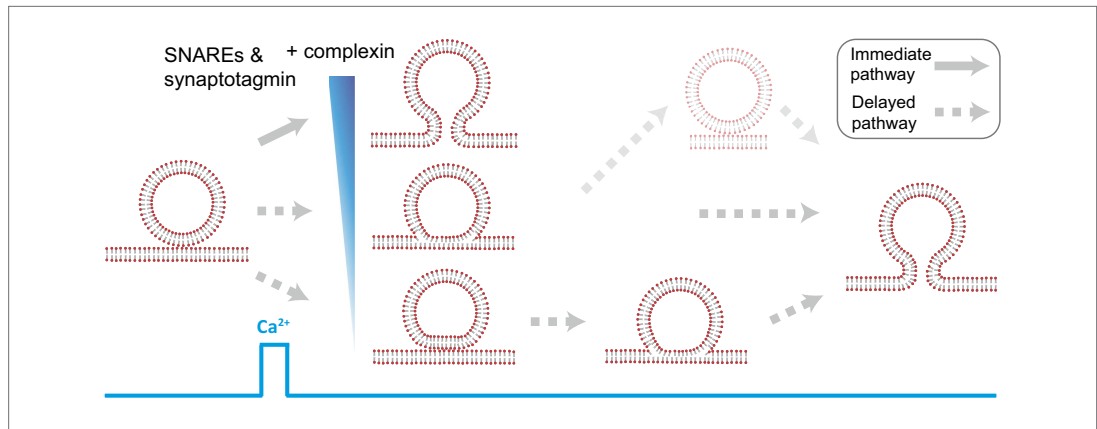

**Figure 7**. Model of multiple fusion pathways upon Ca²⁺ injection starting from a point contact. Complexin synchronizes Ca²⁺-triggered fusion by increasing the number of immediate fusion processes from point contact to fusion pore opening, relative to delayed fusion pathways involving stable hemifusion diaphragms and other long-lived intermediates.

intermediates on the 100-ms time scale. This pathway is based on: (1) observation of only two membrane interface classes prior to Ca²⁺-injection by cryo-EM, point contacts and hemifusion diaphragms (*Figure 2H*). (2) Single vesicle–vesicle fluorescence intensity time traces that recorded both the initial state of the system as well as the temporal sequence of events upon Ca²⁺ injection (*Figure 4A*). We found that all instances of immediate fusion events upon Ca²⁺-triggering started from interacting membranes that were not hemifused. Taken together, it follows that all immediate fusion events started from hemifusion-free point contacts. It is of course possible that this immediate pathway proceeds from point contact to fusion pore via a transient stalk, however, the lifetime of such a transient stalk would have to be faster than the 100-ms time resolution of our instrument. Our experiments definitely rule out the existence of a stable hemifusion intermediate for fast Ca²⁺-triggered fusion.

## Complexin increases the number of immediate fusion events

Our discovery of the immediate fusion pathway (point contact to full fusion) raised the hypothesis that for neurotransmitter release, other synaptic proteins introduce a preference for this immediate pathway upon Ca²⁺-triggering. We confirmed this hypothesis by observing that complexin (in combination with SNAREs and synaptotagmin 1) favors this immediate fusion pathway upon Ca²⁺ triggering, a result that correlates well with complexin's activating role in vivo for synchronous release upon an action potential (*Maximov et al., 2009*). Specifically, we observed that complexin significantly increased the number of immediate fusion events that occur right upon Ca²⁺-injection (*Figure 6*). This effect is especially pronounced at the lowest Ca²⁺ concentration that we tested (250 µM). Thus, complexin suppresses the emergence of long-lived hemifusion intermediates, effectively leading to a higher probability of immediate fusion upon Ca²⁺-triggering. Moreover, complexin was recently found to prevent spontaneous fusion in the absence of Ca²⁺ in another model system with proteoliposomes (*Malsam et al., 2012*).

There are two distinct N-terminal sequence elements that required for complexin's activating role in vivo, one of which (the accessory α-helix) clamps SNARE complexes (*Krishnakumar et al., 2011*). Our results suggest a possible explanation activating function of complexin by imposing geometric restraints on *trans* SNARE complexes through inter-SNARE complex interactions (*Kümmel et al., 2011*), and thereby possibly preventing the formation of long-lived hemifusion intermediates upon Ca²⁺-triggering. Of course, other explanations are also possible, and such molecular mechanisms could be tested in the future by combining single vesicle–vesicle fusion experiments with single molecule observations.

## Selection of the immediate pathway by synaptic proteins

A recent study showed that SNAREs alone can produce a variety of spontaneous (i.e., without Ca²⁺-triggering) fusion pathways on the minute time scale at elevated temperature (30°C) (*Hernandez et al., 2012*), including point contacts, as well as extended close contacts, and hemifusion diagrams. Thus,

biological membranes are poised to undergo fusion via a number of different pathways once they are brought into close proximity by the action of SNAREs. This variety of pathways could have provided a 'noisy' background for evolutionary selection of the immediate pathway that we discovered here. Factors such as complexin may have evolved to select this immediate pathway out of all possible pathways, and may have offered a distinct advantage for fast $Ca^{2+}$-evoked release and efficient communication between neurons.

## Materials and methods

### Protein expression and purification

Full-length rat proteins of syntaxin 1A, SNAP-25A, synaptobrevin 2, synaptotagmin 1, and complexin 1 were expressed and purified essentially as described previously (*Kyoung et al., 2011*). In order to achieve higher yield and purity for the single vesicle–vesicle microscopy experiments, we applied considerable improvements as detailed below. As before, we used a cysteine-free mutant of SNAP-25A (C84S, C85S, C90S, and C92S), and the single site mutants of syntaxin (S193C) and synaptobrevin (S28C); the latter mutants offered the option for fluorophore labeling and single molecule number density experiments (*Kyoung et al., 2011*). As before, full-length synaptotagmin 1 was expressed in *Sf9* insect cells (Invitrogen, Grand Island, NY), purified by $Ni^{2+}$-nitrilotriacetic acid (NTA) sepharose (Qiagen, Hilden, Germany) affinity chromatography, followed by his-tag cleavage, size exclusion chromatography, and, finally, ion exchange chromatography.

SNAP-25 was expressed with an N-terminal TEV cleavable his-tag from plasmid pTEV5 (*Rocco et al., 2008*) in BL21 (DE3) *Escherichia coli* cells (Novagen, EMD Chemicals, Gibbstown, NJ) and purified by $Ni^{2+}$-NTA sepharose affinity chromatography. After removal of the his-tag by overnight cleavage with TEV protease, the sample was further purified by size exclusion chromatography using a Superdex 200 10/300 column (GE Healthcare, Uppsala, Sweden) in buffer containing 20 mM HEPES, pH 7.5, 100 mM NaCl, and 4 mM dithiothreitol (DTT).

Complexin was expressed as a his-tagged protein from vector pET28a (Novagen, EMD Chemicals, Gibbstown, NJ, USA) in BL21 (DE3). Overnight cultures (4 l) were grown in autoinducing LB medium (*Studier, 2005*) at 30°C, harvested, lysed, and then purified using the protocol described in reference (*Kyoung et al., 2011*).

Full-length rat syntaxin and synaptobrevin were expressed with a N-terminal, TEV protease cleavable, hexa-histidine tag from plasmid pTEV5 (*Rocco et al., 2008*). Proteins were expressed overnight at 25°C in autoinducing media (*Studier, 2005*) in strain C43 (*Miroux and Walker, 1996*). Cell pellets from a 8 l of culture were suspended in 400 ml of 50 mM NaPi pH 8, 1 M NaCl, 5 mM EDTA, and 1 mM PMSF supplemented with Complete Protease Inhibitor Cocktail tablets (Roche, Basel, Switzerland), and broken by three passes through a M-110-EH microfluidizer (Mircrofluidics Corp., Newton, MA) at 15,000 PSI. Inclusion bodies were removed by two consecutive 10 min 10,000 RPM spins in a JA-14 (Beckman Coulter, Brea, CA) rotor, and the membrane fraction collected by centrifugation at 40,000 RPM for 2 hr in a Ti-45 (Beckman Coulter, Brea, CA) rotor. Membranes containing syntaxin were washed with 10 mM Tris-$H_2SO_4$, pH 7.5, 10 mM EDTA, 10% (wt/vol) glycerol, centrifuged at 40,000 RPM for 1.5 hr in a Ti-45 rotor, the pellet resuspended in 20 mM HEPES, pH 7.5, 500 mM NaCl, and 1 mM *tris*(2-carboxyethyl)phosphine (TCEP), and centrifuged for an additional 1.5 hr in the same rotor. Membranes were suspended to a concentration of 5 mg/ml in 20 mM HEPES, pH 7.5, 500 mM NaCl, 1 mM TCEP, 10 mM imidazole, 1 mM PMSF and EDTA-free Complete Protease Inhibitor Cocktail (Roche, Basel, Switzerland). Dodecylmaltoside was added to 2%, and after incubation at 4°C for 1 hr, the sample was centrifuged for 35 min at 55,000 RPM in a Ti-70 (Beckman Coulter, Brea, CA) rotor, and the supernatant loaded onto a 0.75 ml column of Ni-NTA agarose (Qiagen, Hilden, Germany). The column was washed with 20 mM HEPES, pH 7.5, 300 mM NaCl, 1 mM TCEP, 20 mM imidazole, 110 mM octylglucoside (OG), and the protein eluted in that buffer containing 500 mM imidazole and 1 M NaCl. 1 mM EDTA was immediately added to each fraction, and those fractions containing protein were loaded onto a Superdex 200 HR 10/30 (GE Healthcare, Uppsala, Sweden) that was equilibrated with 20 mM HEPES, pH 7.5, 300 mM NaCl, 1 mM TCEP, 110 mM OG. Protein fractions were pooled, and digested with TEV protease for 1 hr at ambient temperature, after which the reaction was complete and the TEV protease had precipitated. TEV was removed by centrifugation at 5000 RPM in an Eppendorf (Hamburg, Germany) model 5804 R tabletop centrifuge.

All proteins were reconstituted into vesicles within 1 to 2 hr after the final purification steps in order to prevent potential degradation and aggregation.

## Vesicle reconstitution of full-length syntaxin, synaptobrevin, and synaptotagmin 1

Proteins were reconstituted into acceptor (syntaxin/SNAP-25) and donor (synaptobrevin/synaptotagmin 1) vesicles with a detergent depletion method as previously described (*Kyoung et al., 2011*; *Kyoung et al., 2012*). Briefly, lipid films and other membrane components with compositions that mimic synaptic vesicle and active zone membranes were dissolved in 110 mM OG buffer and purified proteins (synaptobrevin/synaptotagmin 1 and syntaxin for donor and acceptor vesicles, respectively) were added. The synaptobrevin and syntaxin protein to lipid ratio was 1:200; the synaptotagmin 1 to synaptobrevin molar ratio was 1:4.6, consistent with observations of purified synaptic vesicles (*Takamori et al., 2006*). For acceptor vesicles, SNAP-25 solution (five times the concentration of syntaxin) was added to the protein–lipid mixture in order to prevent formation of 'dead-end' 2:1 syntaxin/SNAP-25 complexes. 3.5 mol% phosphatidylinositol 4,5-bisphosphate (PIP$_2$) was added to the acceptor vesicles only; this concentration is within the range of that observed in the plasma membrane of PC12 cells (*van den Bogaart et al., 2011a*). Detergent free buffer (20 mM HEPES, pH 7.4, 90 mM NaCl, 1% 2-mercaptoethanol) was added to the protein–lipid mixture until the detergent concentration was at the critical micelle concentration and the solutions were purified with a CL4B column and dialyzed overnight with Bio-beads SM2 (Bio-rad, Hercules, CA) in detergent-free 'Vesicle Buffer' (20 mM HEPES, pH 7.4, 90 mM NaCl, 20 μM EGTA, 1% 2-mercaptoethanol). Donor vesicles were labeled with DiD and formed in the presence of 50 mM sulforhodamine B (Invitrogen, Grand Island, NY), prior to size exclusion chromatography and dialysis. Reconstituted vesicles were used within 1 to 2 days for optical microscopy and cryo-EM experiments.

The homogeneity of the reconstituted donor and acceptor vesicles obtained from our reconstitution protocol was extensively tested with cryo-EM, light scattering, single molecule counting experiments in order to determine the protein number distributions in single vesicles, SDS gel electrophoresis of reconstituted vesicles, and determination of the directionality of reconstituted proteins as previously described (*Kyoung et al., 2011*). Furthermore, in this work, for each experimental condition we performed multiple experiments from different protein preparations and reconstitutions to ensure that the results are not dependent on a particular preparation.

## Shortcomings of lipid-mixing-only assays

In vitro ensemble assays with reconstituted proteoliposomes have been widely used to study SNARE-mediated membrane fusion (*Weber et al., 1998*). They were important to establish that SNAREs have some fusogenic activity. However, nearly all of these studies monitored lipid mixing, that is the exchange of lipids between membranes. Despite the ease of such ensemble lipid mixing assays, conclusions drawn in the absence of content mixing indicators were often incomplete, or in some cases, they did not correlate well with physiological observations (*Sørensen, 2009*). This is not surprising since lipid mixing is necessary for content mixing, but it is not sufficient (compare *Figures 6 and 8*).

Content mixing measurements have been notoriously difficult to achieve for ensemble-based assays because of a number of technical hurdles, including potential leakiness of proteoliposomes, aggregation, and vesicle rupture that may plague ensemble experiments. Such phenomena may produce a large fluorescence intensity change that cannot easily be distinguished from genuine fusion. Another issue with commonly used lipid-mixing ensemble experiments is that they cannot distinguish between docking and hemifusion/fusion since the observed lipid-mixing signal depends on both processes (*Cypionka et al., 2009*). Even measuring content mixing in an ensemble experiment would not be able to discern effects caused by differences in docking and fusion since the ensemble-average fluorescence signal depends on both docking and complete fusion. On a different note, some of the first ensemble lipid mixing studies used an unreasonably high protein to lipid ratio (e.g., 1:10 for synaptobrevin-reconstituted vesicles [*Weber et al., 1998*]); this is a concern since high protein concentrations are known to cause vesicle instabilities. Finally, ensemble measurements may obscure heterogeneous fusion pathways since they only observe averages rather than individual fusion events. This is indeed a problem for ensemble experiments since single-vesicle lipid-mixing experiments revealed multiple intermediates in SNARE-mediated fusion (*Yoon et al., 2008*; *Karatekin et al., 2010*), and the single vesicle–vesicle content/lipid mixing results presented in this work uncovered heterogeneous fusion pathways for Ca$^{2+}$-triggered fusion in the presence SNAREs, synaptotagmin, and complexin.

Even at a single vesicle level, lipid mixing is not necessarily indicative of content mixing, especially since it is often not possible to resolve two distinct lipid mixing events that would correspond to outer and inner leaflet mixing of a single interacting vesicle pair, respectively. Furthermore, many of these

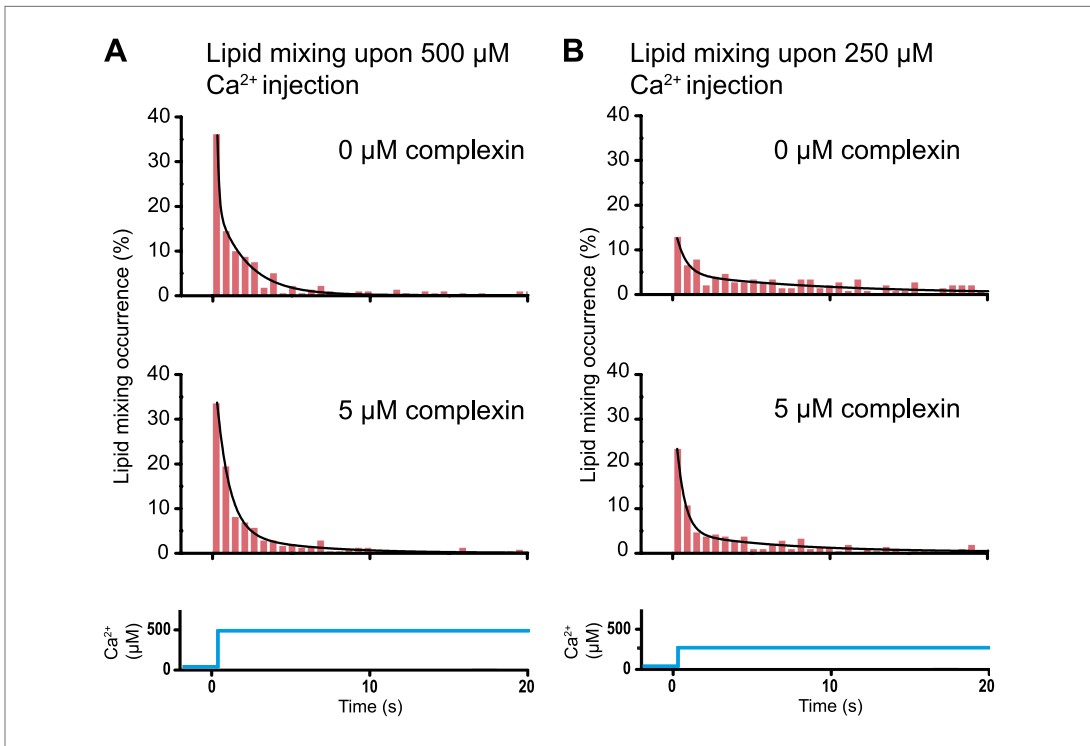

**Figure 8**. Histograms of the occurrence of $Ca^{2+}$-triggered lipid-mixing events with and without 5 μM complexin, corresponding to the experiments shown in **Figure 6**. (**A**) $Ca^{2+}$ injection at 500 μM. (**B**) $Ca^{2+}$ injection at 250 μM. Black lines are fits to exponential decay functions over the entire 50 s. For 500 μM $Ca^{2+}$, in the absence of complexin the fitted function is $f(t) = 0.0014 + 2.66e^{-t/0.11} + 0.22e^{-t/1.94}$ and it is $10^{18}$ times more likely compared to a single exponential fit, and in the presence of 5 μM complexin the fitted function is $f(t) = 0.0006 + 0.42e^{-t/0.81} + 0.05e^{-t/4.76}$ and it is $10^{10}$ times more likely compared to a single exponential fit. For 250 μM $Ca^{2+}$, in the absence of complexin the fitted function is $f(t) = 0.0023 + 0.13e^{-t/0.61} + 0.05e^{-t/9.05}$ and it is $10^6$ times more likely compared to a single exponential fit, and in the presence of 5 μM complexin the fitted function is $f(t) = 0.003 + 0.33e^{-t/0.51} + 0.05e^{-t/6.34}$ and it is $10^{14}$ times more likely compared to a single exponential fit.

experiments were typically performed on mixtures of donor/acceptor vesicles in solution at a constant $Ca^{2+}$ concentration, rather than starting from the metastable state of interacting vesicle pairs at zero $Ca^{2+}$ used in our system (see below). This introduces the possibility of synaptotagmin C2 *cis*-interactions with its own membrane when $Ca^{2+}$ is present prior to docking of vesicles (**Vennekate et al., 2012**), and thereby reducing synaptotagmin *trans*-interactions with the acceptor vesicles. The single vesicle lipid mixing (that is, without content monitoring) experiments by (**Lee et al., 2010**) are a case in point. In their experiments a paradoxical 'decrease' of $Ca^{2+}$-triggered lipid mixing from 10 to 100 μM was observed, assuming background level at 100 μM. In contrast, our system exhibits the more expected increase in fusion (content mixing) as the $Ca^{2+}$ concentration is increased (**Figure 6**). A more detailed comparison of single vesicle systems is available in (**Kyoung et al., 2012**).

## Single vesicle–vesicle content/lipid mixing system

To overcome the shortcomings of both ensemble and single-vesicle lipid-mixing-only assays, we developed a single vesicle–vesicle method that simultaneously monitors the temporal sequence of changes in both lipid and content mixing using characteristic dequenching 'jumps' of two spectrally distinct fluorescent dyes, DiD and sulforhodamine B, respectively (**Kyoung et al., 2011**; **Kyoung et al., 2012**). The size of the soluble content dye sulforhodamine B is only slightly larger to that of certain neurotransmitters, such as glutamate or serotonin. For the content dye, dequenching may occur by content mixing, leakage, or photobleaching. Complete fusion is characterized by a rapid jump in content dye fluorescence followed by relatively steady fluorescence intensity. Leakage leads to a quick spike,

followed by rapid disappearance of fluorescence intensity. Photobleaching leads to slow increases and then decreases of fluorescence intensity. Thus, events that do not lead to content mixing can be easily distinguished from leakage and photobleaching for each individual vesicle pair.

Our single vesicle–vesicle system starts from a metastable state of docked vesicle pairs that is established by an incubation period at zero $Ca^{2+}$ concentration. The incubation period at zero $Ca^{2+}$ is followed by $Ca^{2+}$ injection at a defined concentration. Our system discriminates between docking and fusion since, by definition, we only monitor lipid and content mixing for vesicles that are interacting, that is, docked vesicles. During the incubation period at zero $Ca^{2+}$ concentration, some vesicles already undergo hemifusion, but a sizeable population (typically, more than 50%) will be available after the incubation period that has not yet hemifused or fused (*Figures 2 and 4*).

Extensive characterizations and controls were performed to validate our system (*Kyoung et al., 2011*). To our knowledge this is the only in vitro system that monitors the temporal sequence of both content and lipid mixing events between single vesicle pairs on a 100-ms time scale, differentiates between docking, hemifusion, complete fusion, and vesicle bursting, that triggers the fusion process by injection of $Ca^{2+}$ starting from a metastable state of interacting vesicle pairs that have been incubated at zero $Ca^{2+}$, and that allows the addition of other factors during the incubation and observation stages.

In our initial studies we applied a relatively high $Ca^{2+}$-concentration in order to test our system under a variety of conditions (*Kyoung et al., 2011*). This was a reasonable strategy at an early stage of technique development and proof-of-principle studies. After improvements in optical instrumentation (see below) as well as protein expression and purification (described above), we are now able to routinely perform experiments at 250–500 µM $Ca^{2+}$.

## Improvements to the single vesicle–vesicle content/lipid mixing system

For the experiments described in this work, we used a prism type total internal reflection (TIR) setup in order to maximize the useable field of view. In addition, we employed one excitation wavelength (532 nm) to excite both content and lipid dyes simultaneously rather than using two lasers as in our previous setup. This approach produced a better signal-to-noise ratio of the content dye fluorescence intensity time traces, and reduced photobleaching of the lipid dyes.

The surface preparation, acceptor vesicle immobilization, and donor vesicle incubation were performed essentially as previously described (*Kyoung et al., 2011*; *Kyoung et al., 2012*) with modifications as detailed here. A PEG-coated glass surface was prepared as described in *Diao et al., 2012* and then incubated with a neutravidin solution (pH 7.5, 50 µg/ml, 20 mM HEPES, 90 mM NaCl) for 10 min followed by a buffer wash in order to prevent non-specific surface binding. Control experiments with acceptor-free PEG-coated surface were performed in order to ensure that non-specific binding of donor vesicles was rare; furthermore, we had previously performed a control experiment with a soluble synaptobrevin fragment to disrupt SNARE complex formation which resulted in loss of donor vesicle binding to surface-tethered acceptor vesicles (*Kyoung et al., 2011*). The effect of surface tethering on vesicle interactions and fusion has been shown to be negligible (see Figure S7 in [*Yoon et al., 2006*] and Figure 3 in [*Diao et al., 2010*]).

Unlabeled acceptor (syntaxin/SNAP-25) vesicles were immobilized on the modified surface, and excess vesicles were removed by thoroughly washing with Vesicle Buffer. Donor vesicles (approximately 500× dilution) encapsulating sulforhodamine B and labeled with DiD were introduced into the sample chamber. After 20–30 min incubation, excess vesicles were removed by extensive washing with Vesicle Buffer followed by $Ca^{2+}$ injection using a motorized syringe pump with a flow rate of 33 µl/s.

For the experiments that included complexin, 5 µM complexin and approximately 500× diluted donor vesicles were incubated at ambient temperature for 20–30 min followed by a Vesicle Buffer wash in the presence of complexin.

All experiments were performed at ambient temperature. It should be noted that experiments in live neurons by photolysis of caged $Ca^{2+}$ compounds were also carried out at ambient temperatures (*Schneggenburger and Neher, 2000*; *Sun et al., 2007*). In principle, our system is suitable for studies at other temperatures as well, although the stability of the vesicles would have to be carefully checked at elevated temperatures.

## Extension of the vesicle–vesicle microscopy system to determine the initial state of the membrane interface

For the experiments shown in *Figure 4* we performed fluorescence intensity acquisition right at the start of the experiment. We determined the state of the membrane interface (docked or hemifused) right before $Ca^{2+}$ injection by assessing if changes occurred during the incubation period (laser

illumination was interrupted during the incubation period to prevent photobleaching). We injected donor vesicles (at 50× dilution) for 30 s followed by a buffer wash, and then immediately recorded the fluorescence intensities of the lipid dyes and content dyes for short periods (1–4 s). A 20 min period followed without laser illumination to prevent photobleaching. The laser illumination resumed 5 s prior to injection of $Ca^{2+}$ into the sample chamber. Since the sample remained on the microscope stage during the 20 min incubation period, the positions of the docked vesicles remained unchanged.

### Analysis of the content and lipid dye fluorescence intensity traces

Content and lipid dye fluorescence intensity time traces from individual single vesicle pairs were analyzed using the single molecule software developed in Dr. Taekjip Ha's lab at University of Illinois (*Diao et al., 2012*). A 200 ms time binning was used for recording and a 600 ms moving average was used for data analysis.

### Definition of jumps in fluorescence intensity time traces

The lipid and content mixing events are characterized by significant fluorescent intensity jumps in their respective recorded time traces (see representative examples in *Figures 1B and 4*). Typically five to ten time points immediately before and after the jump were used to evaluate the fluorescence intensity change as well as the noise level. A jump was considered 'significant' if the change in fluorescence intensity ΔI was greater than the average noise level $\sigma = \sqrt{(\sigma_1^2 + \sigma_2^2)}$ where $\sigma_1$ and $\sigma_2$ are the standard deviations of I before and after the jump. For the data used in this work, the typical signal to noise ratio (SNR = ΔI/σ) was in the range 4 to 10.

### Generation of content and lipid mixing histograms

For the histograms shown in *Figures 6 and 8*, time differences between $Ca^{2+}$ injection and instances of lipid mixing and content mixing were determined by inspection of the individual time traces; the time stamp of $Ca^{2+}$ injection was defined as the instance of the first lipid-mixing event among all docked vesicles. Histograms of these time differences were generated for lipid and content mixing instances, respectively. Histograms were normalized with respect to the number of docked vesicles that exhibited at least one lipid dye fluorescence intensity jump during the observation period. Histograms were fitted to decay functions with one exponential and sum of two exponentials, respectively, and the fit used that is more likely correct based on the Akaike information criterion implemented in OriginPro 8.6 (OriginLab, Inc.).

For the experiments shown in *Figures 6 and 8* at 500 μM $Ca^{2+}$-injection in the absence of complexin we observed 245 lipid mixing events and 106 content mixing events (out of a total of 1794 spots), at 500 μM $Ca^{2+}$-injection in the presence of 5 μM complexin we observed 248 lipid mixing events and 150 content mixing events (out of a total of 1852 spots), at 250 μM $Ca^{2+}$-injection in the absence of complexin we observed 157 lipid mixing events and 55 content mixing events (out of a total of 3085 spots), and at 250 μM $Ca^{2+}$-injection in the presence of 5 μM complexin we observed 215 lipid mixing events and 100 content mixing events (out of a total of 3726 spots). The histograms are combinations of a total of five independent experiments for each of the four conditions.

### Cryo-EM

Frozen-hydrated samples of SNARE-containing vesicles prior to, and approximately 35 s after, $Ca^{2+}$ addition were prepared using the procedures for observation in cryo-EM as described previously (*Kyoung et al., 2011*). Briefly, samples were observed by cryo-EM in low dose conditions using a CM200F electron microscope (FEI, Hillsboro, OR) operating at 200 kV. Images of both conditions were collected at a 50,000× magnification and 1.5 μm under-focus on a 2k × 2k Gatan UltraScan 1000 camera (Gatan Inc., Pleasanton, CA).

We observed many vesicle–vesicle interactions that were away from the holey carbon grid (*Figure 2*). Cryo-EM images of individual donor and acceptor vesicle populations (in Supplementary figure S3, panels A, B in [*Kyoung et al., 2011*]) showed no pronounced interactions between same vesicles, so the grid has a negligible effect, if any, on vesicle interface formation.

Analyses were performed from a total of 16 (without $Ca^{2+}$) and 21 (with $Ca^{2+}$) cryo-EM micrographs with good contrast (selected images are shown in *Figure 2* and all raw images are available in *Figure 2—source data 1*).

### Measurement of vesicle diameters

Vesicle diameters were marked with two points for each vesicle using the 'boxer' feature in EMAN (*Ludtke et al., 1999*). Diameters in nm were calculated from the pairs of coordinates using Spider

(*Frank et al., 1996*). The vesicle diameter distribution histograms (*Figure 2G*) were calculated with Matlab (MathWorks, Inc.) using 10 nm binning for all vesicles for which an entire section was visible in a particular image (a total of 228 and 237 vesicles in the images at zero $Ca^{2+}$ and after $Ca^{2+}$ addition, respectively). A Student's t-test was performed to assess the significance of the increase of the mean of the vesicle diameter distribution after $Ca^{2+}$ addition.

## Definitions used to classify vesicle interfaces observed in cryo-EM images

In the observed cryo-EM images the most prominent features of lipid membranes are the electron-dense head groups, rich in phosphorous atoms, which appear as high contrast dark lines when observed on end (along the direction of the electron beam—see for example [*Tahara and Fujiyoshi, 1994*; *Lambert et al., 1997*]). A liposome bilayer membrane appears as two parallel black lines at its circumference. Hence, contacts between liposomes can be classified depending on the number of dark lines and their respective distance at the interface. All types of interfaces in our data could be divided into four classes, defined as follows:

### Point contact
Close apposition between two vesicles with a distance of 1–5 nm between bilayers, without merging or membrane deformation. The observed interface was typically ≤10 nm (along the membrane direction).

### Extended close contact
Contact characterized by a triple-line feature over distances ≥10 nm along the interface, with a denser (darker) middle line suggesting a close (<1 nm) double-bilayer contact, accompanied by membrane deformation around the interface (flattening).

### Hemifusion diaphragm
Extended (>10 nm along membrane) double-layer (two lines) contact between two vesicles, accompanied by flattening at the interface.

### Mixed interface
Transition between an extended close contact (characterized by three lines) and a hemifusion diaphragm (characterized by two lines) within a single vesicle–vesicle interface.

For the quantitative interface analysis (*Figure 2H*), the interfaces were selected individually using 'boxer' in all good contrast micrographs showing clearly the lipid bilayers and interfaces. Interfaces were measured and selected according to the above classifications from cryo-EM images of both conditions (before and after $Ca^{2+}$ addition) using the 'boxer' feature in EMAN (*Ludtke et al., 1999*), and their number calculated separately for both conditions. The analysis was performed from a total of 78 and 122 vesicle–vesicle interfaces before and after $Ca^{2+}$ addition, respectively. The bilayer–bilayer distances were measured using data processing in Spider (*Frank et al., 1996*).

## Density profile analysis of cryo-EM images

For image density measurements, small 80 × 80 pixel images of vesicle interface regions were selected and cut out of micrographs using the 'boxer' feature in EMAN (*Ludtke et al., 1999*), then processed in Spider (*Frank et al., 1996*) in order to generate average density profiles at selected points along the interface. Briefly, images were rotated so that the lipid bilayers at the interface were aligned with the Y axis, the contrast was flattened (using ramp removal) and inverted, and then average density was calculated over 10 pixel lines, corresponding to 2-nm thick sections perpendicular to the interface, to improve the signal to noise ratio. Average densities were plotted out as function of distance (nm) to measure the separation between lipid layers (maxima in density plots) of the vesicle interface regions.

## Overlay of models of the complex between synaptotagmin 1 and the SNARE complex

For the models presented in *Figures 5B,C*, we assumed that the significant non-bilayer density feature corresponds to the C2 domains of synaptotagmin 1, that the $Ca^{2+}$ binding loops of synaptotagmin 1 are close to the bilayer, that the C-terminal end of the first α-helix of SNAP-25 is close to bilayer in order to allow interaction of the adjacent palmitoylated cysteine residues with the membrane, and that the transmembrane domains of syntaxin and synaptobrevin are close to the point contact between

vesicles. These assumptions produced a reasonable overlay with the cryo-EM image. For the model shown in *Figure 5B*, we estimated that four α-helical turns would be able to bridge a 3.0 nm distance ($R_{rms}$) in a random coil conformation, using a persistence length $l_p$ of 0.9 based on single molecule FRET experiments of synaptobrevin (*Choi et al., 2011*) ($R_{rms}^2 = 2l_p n \times 0.36$ nm, where *n* is the number of residues which is 14 for two heptad coiled coil repeats). Thus, the last four C-terminal helical turns of synaptobrevin were modeled as a random coil in order to approximately account for the distance that a portion of the synaptobrevin polypeptide chain would have to extend in order to reach the vesicle membrane. For the model shown in *Figure 5C*, six C-terminal α-helical turns of synaptobrevin and one α-helical turn of syntaxin were modeled as a random coil in order to approximately account for the required extended linkers to the transmembrane domains.

## Acknowledgements

We thank Drs. James Rothman, Reinhard Jahn, and Alex Hoepker for stimulating discussions, Joseph Rizo for providing the coordinates of the NMR-derived model of the SNARE/synaptotagmin complex, Dr. Taekjip Ha for single molecule software, and Drs. Chirlmin Joo and Minglei Zhao for figure preparation.

## Additional information

### Competing interests
ATB: Reviewing Editor, *eLife*. The remaining authors have no competing interests to declare.

### Funding

| Funder | Grant reference number | Author |
|---|---|---|
| Howard Hughes Medical Institute | | Axel T Brunger, Eva Nogales |
| National Institutes of Health | R37-MH63105 | Axel T Brunger |

The funders had no role in study design, data collection and interpretation, or the decision to submit the work for publication.

### Author contributions
JD, Conception and design, Acquisition of data, Analysis and interpretation of data, Drafting or revising the article; PG, Acquisition of data, Analysis and interpretation of data, Drafting or revising the article; DJC, Conception and design, Contributed unpublished essential data or reagents; MK, Acquisition of data; YZ, Analysis and interpretation of data; SS, Contributed unpublished essential data or reagents; AN, Contributed unpublished essential data or reagents; MP, Contributed unpublished essential data or reagents; AS, Contributed unpublished essential data or reagents; MV, Contributed unpublished essential data or reagents; ASH, Contributed unpublished essential data or reagents; EN, Drafting or revising the article; SC, Drafting or revising the article; ATB, Conception and design, Analysis and interpretation of data, Drafting or revising the article, Contributed unpublished essential data or reagents

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
