## [Decision Letter]

Thank you for choosing to send your work entitled “Synaptic proteins promote calcium-triggered fast transition from point contact to full fusion” for consideration at *eLife*. Your article has been evaluated by a Senior Editor and 3 reviewers, one of whom is a member of *eLife's* Board of Reviewing Editors. The following individuals responsible for the peer review of your submission want to reveal their identity: Jose Rizo-Ray and James Rothman.

The Reviewing Editor and the other reviewers discussed their comments before we reached this decision, and the Reviewing Editor has assembled the following comments based on the reviewers' reports. Our goal is to provide the essential revision requirements as a single set of instructions, so that you have a clear view of the revisions that are necessary for us to publish your work.

**General assessment:**

The reviewers find that the paper describes a very interesting study of the mechanism of membrane fusion induced by the neuronal SNAREs syntaxin-1, synaptobrevin and SNAP-25, together with the Ca^2+^ sensor synaptotagmin-1 and the SNARE complex-binding protein complexin.

**Central conclusions:**

1. Calcium-triggered immediate fusion starts from a membrane-membrane point-contact and proceeds to complete fusion within “milliseconds”, without discernible intermediates.

2. Pathways that involve a stable hemifusion diaphragm only result in fusion after seconds.

3. Complexin shifts the fusion pathways towards the fast pathway.

4. Assays that only measure membrane component mixing, without content mixing measurements, may give misleading results.

The idea of multiple fusion pathways, fast fusion events with no discernible intermediates, and the slow hemi-fusion driven events is a novel concept. The strength of this work is the single vesicle-vesicle system, which is now functional under more physiological levels of Ca^2+^ and provides a level of resolution that is unparalleled. The combination of the fusion assay with the cryoEM studies is a noted strength of the study.

The effect of CPX in increasing the probability of the immediate fusion events is in line with recent findings on the role of the CPX in synchronizing the Ca^2+^-triggered fusion, and quite nicely explains how the zig zag array in addition to clamping fusion might aid in the enhancing the rate of fusion itself and is quite appealing, but this could be explained more clearly in the manuscript.

**Required revisions:**

The reviewers raise a number of concerns that must be adequately addressed before the paper can be accepted. Some of the required revisions will likely require further experimentation within the framework of the presented studies and techniques.

1. An essential assumption for all of the work is that the two types of reconstituted vesicles (donor and acceptor) are homogeneous populations, all with the same make up of lipids and proteins. Significant heterogeneity in terms of protein constituents would completely invalidate the conclusions. Convincing evidence of high homogeneity, or a reference to a convincing study showing such, must be included.

2. The results are for the most part presented in a “semiquantitative” manner, without proper attention to definition of quantitative criteria for classification into the various classes of contacts in the EM and release patterns in the optical measurements. Additionally some of the conclusions are based only on very simple measurements or visual inspection of data, particularly the histograms in Figures 6 and 8. Presenting only the parameters of exponential fits is not adequate. Further treatment of the range of acceptable parameters is necessary. There is also inadequate statistical treatment of the data, even though the term "significant: is often used to describe differences between data from different treatments.

3. The histograms in figure 6, and to a lesser extent figure 8, are very noisy and there are several time bins with no occurrences. The differences are not convincing with only this much data, especially given that the differences between both sets of histograms seem to depend only on the values in the first time bin. These histograms must include a greater number of entries to be convincing.

4. “We performed a quantitative analysis of all observed cryo-EM images” onwards: histograms should be shown for the diameter distributions mentioned, along with a description of classification criteria and statistical analysis.

5. It should be at least mentioned that changes in steady state calcium concentration for many minutes is quite different than the transient increases involved in calcium-dependent transmitter release.

6. The discussion section requires some revision – it is not very accessible and hard to understand at times.

7. Although not necessary, the paper would be greatly strengthened by CryoEM data with CPX.

8. The claimed millisecond time resolution seems inappropriate as the fastest sample interval seems to be 200 ms and 600 ms binning is often used.

9. Figure 4 presents a nice experiment, but numbers of occurrences, classification criteria, and statistics need to be included.

---

## [Author Response]

*1. An essential assumption for all of the work is that the two types of reconstituted vesicles (donor and acceptor) are homogeneous populations, all with the same make up of lipids and proteins. Significant heterogeneity in terms of protein constituents would completely invalidate the conclusions. Convincing evidence of high homogeneity, or a reference to a convincing study showing such, must be included*.

The homogeneity of reconstituted donor and acceptor vesicles obtained from this protocol was extensively tested as previously described (31). Furthermore, for each experimental condition we performed multiple experiments from different protein preparations and reconstitutions to ensure that the results are not dependent on one particular preparation. We have updated the section “Vesicle reconstitution of full-length syntaxin, synaptobrevin, and synaptotagmin 1” in Materials and Methods.

*2. The results are for the most part presented in a “semiquantitative” manner, without proper attention to definition of quantitative criteria for classification into the various classes of contacts in the EM and release patterns in the optical measurements. Additionally some of the conclusions are based only on very simple measurements or visual inspection of data, particularly the histograms in Figures 6 and 8. Presenting only the parameters of exponential fits is not adequate. Further treatment of the range of acceptable parameters is necessary. There is also inadequate statistical treatment of the data, even though the term “significant” is often used to describe differences between data from different treatments*.

We have more clearly defined the criteria to define the various vesicle interfaces in the observed cryo-EM images (section “Definitions used to classify vesicle interfaces observed in cryo-EM images” in Materials and Methods). We have also introduced a new section “Definition of jumps in fluorescence intensity time traces” that describes the statistical criterion to define “significant” jumps in the observed fluorescence intensity time traces. We have now performed fits with two models to all histograms in Figs. 6 and 8: one and a sum of two exponential functions. We have assessed the more likely model with the Akaike information criterion.

*3. The histograms in figure 6, and to a lesser extent figure 8, are very noisy and there are several time bins with no occurrences. The differences are not convincing with only this much data, especially given that the differences between both sets of histograms seem to depend only on the values in the first time bin. These histograms must include a greater number of entries to be convincing*.

We have collected more data for Ca^2+^-injections at both 250 μM and 500 μM. We have updated the histograms in Figures 6 and 8 accordingly.

*4. “We performed a quantitative analysis of all observed cryo-EM images” onwards: histograms should be shown for the diameter distributions mentioned, along with a description of classification criteria and statistical analysis*.

The diameter distributions are now shown in the new panel Fig. 2G. The significance of the increase in vesicle diameters in the distribution is significant as assessed with a Student's t-test to 97% confidence. The description of the classification criteria of the interfaces is now defined in the new section “Definitions used to classify vesicle interfaces observed in cryo-EM images” in Materials and Methods.

*5. It should be at least mentioned that changes in steady state calcium concentration for many minutes is quite different than the transient increases involved in calcium-dependent transmitter release*.

We now more explicitly state in the Introduction that our system mimics Ca^2+^ photolysis experiments in neurons, and that we are working on future extensions of our system to mimic the transient Ca^2+^-increases typical for physiological neurotransmitter release.

*6. Although not necessary, the paper would be greatly strengthened by CryoEM data with CPX*.

We plan to do this in a follow-up study.

*7. The claimed millisecond time resolution seems inappropriate as the fastest sample interval seems to be 200 ms and 600 ms binning is often used*.

We changed the wording to “hundred millisecond timescale” where applicable.

*8. Figure 4 presents a nice experiment, but numbers of occurrences, classification criteria, and statistics need to be included*.

We have supplied the classification in the section “Definition of jumps in fluorescence intensity time traces” in Materials and Methods, and supplied statistics in the caption of Figure 4.